# Understanding Young Taiwanese Consumers’ Acceptance, Sensory Profile, and Drivers of Liking for GABA Oolong Tea Beverages with Cold Infusions

**DOI:** 10.3390/foods11192989

**Published:** 2022-09-26

**Authors:** Mu-Chen Wu, Bo-Kang Liou, Yuh-Shuen Chen, Shih-Chieh Lee, Jia-Jin Xie, Yih-Mon Jaw, Shih-Lun Liu

**Affiliations:** 1Department of Health Business Administration, HungKuang University, Shalu District, Taichung 43302, Taiwan; 2Department of Food Science & Technology, Central Taiwan University of Science and Technology, Beitun District, Taichung 40601, Taiwan; 3Department of Food Science and Technology, HungKuang University, Shalu District, Taichung 43302, Taiwan; 4Department of Medicinal Botanicals and Foods on Health Applications, Da-Yeh University, Dacun, Changhua 515006, Taiwan; 5Bachelor Program for Baking and Beverage, Da-Yeh University, Dacun, Changhua 515006, Taiwan; 6Graduate Institute of Food Culture and Innovation, National Kaohsiung University of Hospitality and Tourism, Xiaogang District, Kaohsiung 812301, Taiwan; 7Department of Chinese Culinary Arts, National Kaohsiung University of Hospitality and Tourism, Xiaogang District, Kaohsiung 812301, Taiwan; 8Department of Nutrition, China Medical University, No. 100, Sec. 1, Jingmao Rd., Beitun District, Taichung 40604, Taiwan

**Keywords:** GABA Oolong tea, cold infusion, CATA, young consumer, acceptance

## Abstract

The sensory qualities of Taiwanese teas are evaluated by the experts from the Tea Research and Extension Station (TRES) at tea competitions held annually. The prices of Taiwanese teas are also influenced by the results of these tea competitions. However, a tea winning an award and having a high sensory quality and price does not mean that it is liked by Taiwanese consumers. The check all that apply method (CATA) is a scientific method of sensory evaluation. It is able to evaluate the sensory characteristics with consumers and is cheap and time-saving. Twelve samples of γ-aminobutyric acid (GABA) Oolong tea made by the Taiwan Tea No. 12 cultivar were selected from the first Taiwanese GABA tea competition in 2020. The aim of this research was to study young Taiwanese consumers’ acceptability for GABA Oolong tea infusions, and their opinions towards the sensory qualities of teas through questionnaires composed of CATA questions and hedonic scales. Based on the results, the CATA method identified 21 important descriptive terms for GABA tea that were selected by over 20% of consumers. It was found that the consumers like GABA Oolong teas with 13 specific sensory characteristics, but dislike the ones with another 6 specific sensory characteristics. We conjecture that the different process of tea production will affect consumers’ preference for GABA Oolong tea. Overall, GABA Oolong tea has the same delightful sensory characteristics as traditional Taiwanese specialty Oolong teas, and is liked by the young Taiwanese consumers.

## 1. Introduction

GABA (γ-Aminobutyric acid) is a non-protein amino acid that comprises four carbon atoms. It commonly exists in a free (ionized) state in the free amino acids of eukaryotes and has a decent hydrophilicity [1]. In medicine, GABA was identified as an important inhibitory neurotransmitter in the central nervous system as early as the 1950s [2]. It is mainly present in the brains of animals and can lower neuronal excitability. GABA contributes to brain activation, increases acetylcholine levels, arouses the parasympathetic nervous system, and promotes the normal functioning of the brain [3]. Many studies on the physiological effects of GABA have highlighted its prominent role in reducing hypertension [4,5], mitigating old age or diabetes-induced depression [6], promoting sleep [7], dispelling the effects of alcohol [8], inducing growth hormone secretion [9], and decreasing the rate of neurodegeneration [10]. A 10% to 15% reduction in blood GABA levels increases one’s likelihood of developing mania, bipolar disorder, anxiety, and depression [5].

In a 1987 study on the metabolism of amino acids in tea trees, the Japanese professor Tsushida accidentally discovered that high amounts of GABA are produced in tea leaves under long-term anaerobic conditions (anoxic fermentation) [11]. This discovery attracted the attention and interest of many Japanese scholars. In the following two years, GABA tea produced under anaerobic conditions became very popular in Japan [12]. It was until early 1994 that this special anaerobic means of GABA tea production was introduced into Taiwan by the associate research fellow Mr. Tsai from the Tea Research and Extension Station (TRES), Council of Agriculture, who was dispatched to Japan to learn about the technique [12].

Most Taiwanese consumers were used to drinking Taiwanese specialty teas with a rich aroma and flavor. Because of the pungent smell produced during the production process, the traditional GABA teas were not liked by consumers [5,12]. After two decades of hard work by the TRES and tea farmers, the optimal method of producing GABA tea was found to be through anaerobic fermentation with two stages of aerobic rolling and three stages of anaerobic fermentation in Oolong tea production, as this method effectively increases the flavor of GABA tea [13]. This unique technique has been used to produce GABA green tea, GABA Oolong tea, and GABA black tea with distinctive flavors and functional benefits [14].

A standard GABA tea contains over 150 mg of GABA in every 100 g of dry tea leaves [12], and this content must be consistent throughout the storage period [5]. The purified GABA has various physiological effects [3,4,5,6,7,8,9,10]. Among the various foods that contain GABA, there is the highest content of GABA in GABA tea. The highest content is up to 300 mg GABA in 100 g tea (on a dry weight basis) [15]. The animal models conducted by Prof. Ōmori have showed the effectiveness of GABA tea at reducing blood pressure [11]. The research team of Prof. Ou also obtained similar results in Taiwan [5]. In addition, GABA tea has healthcare effects such as nerve soothing [16], sleep-promoting [17,18], anti-apoptotic and pro-survival effects [19], prevention of diabetic brain abnormality [20], and lowering blood pressure [15,21,22].

In Taiwan, tea competitions are held annually in tea producing regions, with the winning tea determined by experts from TRES based on the quality of the tea. The price and sales of the award-winning tea will increase significantly. However, the award-winning tea selected by the experts does not reflect that it will be liked and purchased by consumers [23]. In recent years, rapid descriptive analysis has been widely used internationally to evaluate the sensory characteristics of food. Among them, the check all that apply method (CATA) test is the most popular [24,25]. These methods more closely reflect the needs of consumers and differ from the other methods of training professional panelists, which consume a lot of time and money [23,25].

The raw materials used in tea beverages in Taiwan are generally cheaper tea stem and tea fannings. They are the byproducts of tea production, and their quality and sensory characteristics are poor. The tea beverages made with tea stem and tea fannings require additional spices because of the poor flavor of tea infusion [26]. However, various tea beverages made from high quality Taiwanese specialty teas (including black tea, green tea, Oolong tea, etc.) were well-known internationally in recent years [25]. The GABA Oolong tea has additional health benefits and can be used as a raw material for high quality beverages in Taiwan. The price of tea beverages with additional benefits can also be higher than that of regular tea beverages. The tea made from the cultivar, Taiwan Tea Experiment Station No.12 (TTES #12) has a light milk and flower fragrance [27]. It has a unique flavor and is loved by women and young consumers [28]. Taiwanese young generations are very much westernized in their dietary behaviors and choices [23]. Liu et al. (2021) pointed out that cold-brewed infusions of Taiwanese specialty tea have a higher consumer preference level [25].

Therefore, 12 GABA Oolong teas made by the cultivar of TTES #12 from the first GABA tea competition in 2020 in Taiwan were selected as samples for this study. The cold-brewed tea infusions were used for analysis with the CATA method and the acceptance of consumers. The purpose of this study is to understand the sensory profiles, driver of liking, and the acceptance of consumers for GABA Oolong tea among young Taiwanese consumers. This research recruited respondents younger than 30 years old to evaluate the cold tea infusions. These results of this study will be useful for tea farmers and manufacturers to make the GABA Oolong tea which is liked by Taiwanese young consumers.

## 2. Materials and Methods

### 2.1. Tea Sample

A total of 12 tea samples made by TTES #12 cultivar were found among the 41 tea samples entered in the first Taiwanese GABA tea competition in 2020. All 12 Taiwanese GABA Oolong tea samples were selected in this study. TTES #12 is the second largest cultivated variety in Taiwan [27]. The 12 tea samples were produced from the northern (New Taipei City), central (Nantou County) and southern regions (Chiayi County) of Taiwan, and the details are shown in Table 1. The 12 GABA Oolong tea products were tested for pesticide residues prior to consumer testing, and the pesticide residue levels were confirmed to comply with the current regulations in Taiwan. The GABA content of these GABA Oolong tea was tested to meet the commercial standard [5] (150 mg/100 g tea leaves).

For sensory evaluation, the Tea Research and Extension Station, Council of Agriculture, Executive Yuan (TRES) of Taiwan standardized the preparation procedures for both hot and cold tea infusions with tea leaves. The method preparing the cold tea infusions was modified from the method of TRES: to mix 4.5 g of tea leaf and 450 mL room-temperature water in 480 mL plastic jars, and let stand at room temperature for 1 h. After standing, to place in the refrigerator at 2–7 °C for 8 h to extract the infusion [29]. After the leaves are drained, the infusion stored in a refrigerator at 2–7 °C. The evaluation should be completed within 24 h.

### 2.2. Questionnaire Design

The consumer acceptance test and CATA test were the major designs of the questionnaire. A 9-point hedonic scale was incorporated into the questionnaire to investigate the acceptability of consumers including the overall liking, liking of appearance, liking of flavor, liking of texture, and liking of aftertaste. Sensory characteristics of GABA Oolong tea were evaluated by the CATA. A total of 64 attributes of cold-brewed infusion were pre-determined before the test. These mainly refer to the sensory wheel made by the TRES and the attributes defined by the International Sensory Evaluation of Tea and Coffee (ITCE) held by Mr. Yan Jin-Yuan in Taiwan [23]. The basic principle of selecting was to list all the possible sensory attributes about the appearance, aroma, flavor, and taste of Taiwanese tea which could be understood by the Taiwanese consumers.

A total of 62 descriptive terms regarding appearance, basic gustatory and tactility, aroma, flavor, and aftertaste of tea infusions were adopted in the questionnaire of CATA. There were 14 attributes about the appearance, 5 regarding basic gustatory and tactility (6 terms), 18 attributes about the flavor, and 24 attributes about the aftertaste. In the case of aroma, there were 9 different sub-categories in the category of floral flavor including jasmine, osmanthus, white jade orchid, rose, ginger lily, areca flower, lily, violet orchid, and chrysanthemum. If the respondents perceived any applicable attributes in the sub-category of floral flavor from the tea sample, it was considered that the respondents could perceive the floral flavor from the GABA Oolong tea sample. This design helped respondents to easily and clearly perceive the attributes of the GABA Oolong tea. It also avoided the inaccuracy of the CATA test due to the difference of subjective definition of the attributes by each respondent. The different sub-categories of aroma were listed in each category of aroma for the respondents to choose in the questionnaire.

### 2.3. Procedure of Sensory Evaluation

According to Ares et al., 60–80 consumers can be regarded as a reasonable compromise to get a stable sample and descriptor configurations when working with widely different samples [30]. Seven groups of respondents (704 people) participated the evaluation of 41 GABA Oolong tea samples in the first Taiwanese GABA tea competition in 2020. The 12 tea samples made by the TTES #12 cultivar belong to the 41 GABA Oolong tea samples mentioned above and were also evaluated by two of the seven groups of respondents who participated the first Taiwanese GABA tea competition. Two groups of 101 and 100 volunteer respondents were separated to participate in this study, comprising 43 males and 58 females (invited from National Chung Hsing University (NCHU), and Da-Yeh University (DYU)), and 32 males and 68 females (invited from Central Taiwan University of Science and Technology (CTUST)), respectively, in October 2020. The respondents of the first round drank the first 6 tea samples of Table 1, and the respondents of the second round drank the last 6 tea samples of Table 1.

All respondents had to sign a consent form before the test began which included the purpose and time of evaluation and relevant risks to be borne by the respondents.

Before the cold-brewed tea infusions were served to the respondents, they were removed from the refrigerator and stored in the insulation bottle to keep the temperature of infusions between 4 and 7 °C. Before respondents entered the evaluation environment, the sample(s) had been poured into white cups marked by random three-digit numbers. They were presented to the respondents by a sequential monadic technique with William Latin square design. Each Latin square consisted of 6 respondents and 6 samples. Before the experiment, the complete Williams Latin square for 120 respondents was designed by the software Compusense Cloud (Compusense, Inc., Guelph, ON, Canada). The sequences of tea samples for 96 respondents (of the separated 101 and 100 respondents in the first and second rounds) used the duplicate 16 Latin squares. After the respondents entered a quiet classroom in the first round, the 6 tea infusion samples were put in a 30 mL white disposable plastic cup and presented on the pallet at the same time, and the respondents were asked to drink the tea according to the order on the software. However, for the respondents in the second round, tea infusion samples were put in 70 mL white porcelain cups and presented one at a time. Respondents were guided to assess the tea samples arrayed in front of them. Before tasting a new sample, participants had to clean their palates with crackers and pure water. The previous sample could not be retried after evaluation, for the next sample had been started.

For the CATA questions, the respondents only had to check the attributes from the 62 predetermined terms they perceived applicable to the test samples; no consideration of intensity was required. As for the descriptive terms regarding flavor, there were a total of 18 major categories and 100 sub-categories in the questionnaire. When the respondents perceived the attributes of the flavor, whether the attributes of the major category or its sub-categories were perceived. It was deemed that the attributes of the major category were perceived by the respondents. After selection of the attributes for the one sample of CATA test was completed, the 9-point hedonic rating of this sample was assessed by the respondent. The questionnaire of the CATA and 9-point hedonic rating tests were presented to the respondents through the software Compusense Cloud.

### 2.4. Statistical Analysis

XLSTAT [31] was used for the following statistical analyzes. Cochran’s Q test was used to compare the frequencies of attribute selection from the CATA questions, and a post-hoc analysis was conducted using multiple pairwise comparisons with the McNemar test (Bonferroni). The numbers of respondents were not the same between 2 rounds of tests, therefore, the correspondence analysis (CA) with chi-squared distance was calculated on the attributes responded to by over 20% of participants to measure similarities among the tea infusions with 2 different rounds of test. The results of CATA for the first and second round (6 GABA Oolong tea sample each) were analyzed separately. The counts of the numbers of respondents perceived for each sensory attributes of GABA Oolong tea were further converted into a percentage of the perceived respondents to all respondents for each attribute. The converted data of the 2 rounds of CATA for 12 tea samples were analyzed by the covariance model with covariance matrix [32] for principal component analysis (PCA) to compare all the 12 samples at the same time [33]. Consumers’ overall liking of the tea infusions averaged from the 9-point hedonic scores was analyzed by one-way ANOVA in conjunction with Tukey’s honestly significant difference (HSD) test. Finally, partial least squares regression (PLSR) was used for analysis between the hedonic scores and the CATA data with converted data and was performed to understand how the sensory, association, and postprandial perception attributes affected respondents’ hedonic judgments. Similar conclusions from the results of CATA test were drawn by PLSR, PCA, and CA analyses [34,35,36,37,38,39]. Because of the predictive nature of PLSR, it is more appropriate when the aim is to predict liking based on a series of sensory attributes [35]. In this study, the PLSR was used to find the relationship between sensory attributes and overall liking of Taiwanese GABA Oolong teas.

## 3. Results

Most of the respondents were school/institute students because the primary aim of this study was to understanding the acceptance, sensory profile, and drivers of liking for GABA Oolong Tea for young Taiwanese consumers. According to the WHO definition, young people are those aged between 18 and 29 years [40]. There were 75% of young respondents in the first round and 85% in the second round in this study. Regarding their tea drinking habits, 86% of the respondents in the first round and 87% in the second round had the habit of drinking tea. An exceeded 100 respondents were selected to participate in both rounds of the CATA test. It was evaluated from the point of view of statistical sampling distribution, and the resulting data of both rounds were found to be sufficiently representative of the distribution of the population. For the respondents in the first round, 44.6% had heard about GABA Tea and 22.8% had drunk it, and for the respondents in the second round, 45% had heard about GABA Tea and 16% had drunk it (shown in Table 2). The habit of tea drinking has penetrated into all classes of people in Taiwan. As well as traditional hot brewed tea, various new types of cold tea beverages are flourishing in the market in Taiwan. According to the results shown in Table 2, many young consumers had never heard of nor drunk GABA tea in Taiwan. The environment of the three evaluation places was arranged according to the previous study [25]. The places used in the first round were NCHU and DYU; the evaluation is conducted in a place free from interference. The place used for the second round was CTUST; the evaluation was conducted in a professional sensory evaluation room. Each respondent evaluated the tea sample in a separate room.

Table 3 demonstrates the consumers’ acceptability toward Taiwanese GABA Oolong tea, made from the TTES #12 cultivar measured on a 9-point hedonic scale following two rounds of tests. A total of 101 consumers took part in the first round of consumer testing and evaluated six GABA Oolong tea samples; another 100 consumers took part in the second round of testing and evaluated the last six tea samples. These 12 GABA Oolong tea samples were produced by eight different tea manufacturers. We separated two tea samples produced from the same manufacturer into a different round of testing. A total of six tea samples from three manufacturers were implemented according this method. Meanwhile, two tea samples produced from the same company were included in the second round, validating that there were no differences between the evaluation data of the two rounds of consumer tests.

The results show that the sensory acceptability of 12 GABA Oolong tea samples among Taiwanese young consumers ranged from “dislike slightly” to “like slightly”. In comparison with the results obtained by Liu et al. (2021) [25], the acceptability of GABA Oolong tea is similar to Taiwanese specialty teas. After all, young Taiwanese consumers still love and accept the Taiwanese GABA Oolong tea. Table 3 shows that there are no significant differences in consumer acceptability between the different GABA Oolong teas produced by the same manufacturer after two rounds of testing. These findings validate that there are no differences between the results of the two consumer tests. This also shows that the result is objective and credible. It can also be observed from Table 3 that GABA Oolong tea produced by the same manufacturer has similar consumer acceptability.

The last code of the sample code indicates the name of the manufacturer. In terms of the consumers’ overall liking, the GABA Oolong teas made by manufacturers B, J, P, and L were received more favorably by consumers, while those made by manufacturers G and H were received less favorably. The results of consumers’ overall liking were more associated with their preferences regarding flavor-related criteria such as overall flavor, overall taste, and overall aftertaste, instead of the overall appearance (color). This shows that consumers’ choice of GABA Oolong teas is primarily affected by the flavor, taste, and aftertaste of tea.

A comparison of the origin, farming altitude, tea leaf type, and farming method of the 12 GABA Oolong tea samples showed that it is not possible to observe specific trends in sensory attributes under different origins, farming altitudes, tea leaf types, and farming methods of GABA tea. Instead, we found the acceptability of consumers of GABA Oolong tea made from the different manufacturer were quite different, but the ones that were similar were from the same manufacturer. We speculate that the processing procedure affected the sensory attributes and the acceptability of consumers of GABA Oolong tea. However, consumers are less able to distinguish the difference of sensory characteristics due to different raw materials. This hypothesis will be further verified using CATA and other statistical methods.

A study on the consumer acceptability of various sugary hand-shaken tea-based beverages reported that overall acceptability is more closely associated with flavor, taste, and aftertaste than appearance [41]. Our results shows that even though the appearance of GABA Oolong tea is like that of other Taiwanese teas, Taiwanese consumers showed slightly less preference for its flavor, taste, and aftertaste over the other Taiwanese teas.

Figure 1 is a horizontal stacked bar chart showing the overall liking and selection frequency of consumers from two rounds of testing (*n* = 101, 100 for 1st round and 2nd round) for the 12 GABA Oolong tea samples made by the TTES #12. The black figure in the figure indicates the ratio of “Neither like nor dislike”. If this figure is used as the rough boundary standard, the different stripe patterns on the right side of the black figure, respectively, indicate the positive liking ratio of consumers from “like slightly” to “like extremely“, and the left part shows the selection ratio from “dislike slightly” to “dislike extremely”. In Table 3, B-CY-P, B-NP-B2, and B-NP-B1 are the three samples with the highest overall liking, with scores of 5.84, 5.82, and 5.80, respectively. It can be seen from Figure 1 that 59%, 58%, and 61% of consumers have positive comments on these three samples. In Table 3, B-CY-H, B-NT-G2, and B-NT-G1 are the three samples with the lowest overall liking, with scores of 4.44, 4.75, and 4.86, respectively. It can be seen from Figure 1 that 36%, 38%, and 42% of consumers still had positive comments on these three samples. Nearly 40% of consumers still accepted these three tea samples.

As shown in Figure 2 and Figure 3, sensory properties of 12 cold-brewed Taiwanese GABA Oolong tea infusions were identified by respondents’ selection frequencies of the 62 major sensory attributes including 100 sub-categories of flavor. Six separate GABA Oolong tea samples were tested by two different CATA assessments by different respondents. Even though these two sensory evaluations were performed by different respondents, the data of these two tests could be comparable since they were evaluated by the number of respondents essential for obtaining stable sample/descriptor configurations (*n* = 101, 100 for the first round and the second round, respectively) [42]. Many sensory properties were shown to vary similarly among Taiwanese specialty tea, whether they were cold- or hot- brewed. Even consumers can perceive more sensory characteristics of cold-brewed tea infusion [23,25,43].

According to a previous study, attributes could be perceived and recognized by over 20% of respondents as the important characteristics of sample, while the others were irrelevant. Too many irrelevant attributes used in the analysis of CATA will not reflect consumers’ true feelings about the product [25,44,45]. The 26 terms marked with asterisks (*) in Figure 2 were the important attributes for six cold-brewed GABA Oolong tea samples in the first round of testing. In appearance, they include light-yellow color, golden yellow color, clearness, brightness, and lustrousness; in flavor, they include sweetness, sourness, saltiness, bitterness, umami taste, astringency, floral flavor, sugary flavor, honey flavor, fruity flavor, dried fruit flavor, roasted flavor, and woody flavor; in mouthfeel, they include late sweetness, salivating, smoothness, freshness, late fragrance, coolness, lingering taste, and softness. After Cochran’s Q test, there are significant differences (*p* < 0.05) among the six GABA Oolong samples for the 17 sensory characteristics mentioned above. Among them, there are nine sensory characteristics (sourness, saltiness, dried fruit flavor, roasted flavor, woody flavor, late sweetness, salivating, lingering taste, and softness) which are the feelings of consumers among the samples; there is no significant difference (*p* > 0.05). For the item of sour (40–52%; *p* = 0.06), over 50% of consumers checked this sensory characteristic.

The 24 terms marked with asterisks (*) in Figure 3 were the important attributes for six cold-brewed GABA Oolong tea samples in the second round of testing. In appearance, they include light-yellow color, golden yellow color, clearness, brightness, and lustrousness; in flavor, they include sweetness, sourness, saltiness, bitterness, umami taste, astringency, floral flavor, honey flavor, fruity flavor, roasted flavor, and grassy/Chinese herb flavor; in mouthfeel, they include late sweetness, smoothness, freshness, late fragrance, coolness, and softness. After Cochran’s Q test, there are 17 sensory characteristics mentioned above displaying a significant difference (*p* < 0.05) among the six GABA Oolong samples. Among them, there are five sensory characteristics (sourness, roasted flavor, grassy/Chinese herb flavor, late sweetness, and softness) which are the feelings of consumers among the samples; there is no significant difference. For the items of clearness (44–57%; *p* = 0.16), bitterness (55–62%; *p* = 0.65), umami taste (52–61%; *p* = 0.36), astringency (69–79%; *p* = 0.37), and floral flavor (52–69%; *p* = 0.14), over 50% consumers checked these sensory characteristics. This shows that characteristics of sour, clearness, bitterness, umami taste, astringency, and floral flavor are the most obvious sensory characteristic of Taiwanese GABA Tea which are most easily perceived by Taiwanese young consumers.

Comparing the results of both tests, the consumers could not distinguish clearly between the terms related to sugary aromas and fruity aromas, and woody flavor and grassy/Chinese herb flavor. In addition, the results of the two groups of consumers somewhat differed in terms of their perceptions of flavor-related and mouthfeel-related terms.

Based on the comprehensive results of both tests, the major sensory characteristics of Taiwanese GABA Oolong tea perceived by Taiwanese young consumers can be expressed through 21 descriptive terms. Whether significant differences exist between the consumers’ perceptions of the sensory characteristics of the different group of six GABA Oolong tea samples, the findings revealed that 31 and 14 descriptive terms provided, respectively, by the respondent in each of the two tests were used by the consumers to describe the differences between samples as perceived by them. This finding shows that the number of sensory characteristics that determine the distinguishable differences between products varied, with the terminology perceived by over 20% of the consumers. For instance, customers who could distinguish between the differences that exist between products had selected terms such as “light red”; however, in reality, most consumers were indifferent toward these six terms in reality because of the overly low frequency of selection. It can be seen from the results in Figure 2 and Figure 3 that the sensory characteristics of Taiwanese GABA Oolong Tea perceived by the two groups of respondents are similar.

Of the 21 terms commonly selected by the consumers, terms that were selected by consumers for over 50% of their selections were included in the main terminology. Comparatively speaking, the consumers could not distinguish between golden-yellow and light yellow colors of tea liquid, and the consumers in the second test were less able to identify the bitter taste of GABA Oolong tea. Therefore, this study settled on the terminology that resonated with over 20% of consumers to explore the conditions that determine the characteristics of a Taiwanese GABA Oolong tea sample. This approach has been used in many sensory evaluation studies [23,24,25,46,47].

Figure 2 and Figure 3 further show that all consumers in the first test could identify the light yellow and golden yellow colors of GABA Oolong tea. Interestingly, tea samples that most consumers identified as light-yellow (selected by 86 and 71 people, respectively) were identified as golden-yellow by fewer consumers (6 and 29 people, respectively). There were fewer people who identified the color of a tea sample as light-yellow when most identified it as golden-yellow. In the second test, less than 41 consumers could identify golden-yellow-related terms, which validated the aforementioned finding that the consumers could not distinguish clearly between the golden-yellow and light-yellow colors of tea liquid.

When comparing the GABA Oolong tea with higher and lower acceptability according to the results in Table 3, Figure 2 and Figure 3, it was found that consumers have higher overall acceptability to the GABA Oolong tea because of the higher brightness of the color, the lower bitterness and the higher floral, honey, and fruit flavor, and the higher smoothness and softness of the mouthfeel of the tea infusion. If a GABA Oolong tea has a lower honey flavor and softness of the mouthfeel of the tea infusion, the overall acceptability of this GABA tea will be lower.

On the whole, it can be concluded that young consumers in Taiwan can perceive the sensory characteristics of Taiwanese GABA Oolong tea related to the flavor easily, and it is more difficult to perceive the sensory characteristics related to the taste. Compared with traditional Taiwanese GABA tea, GABA Oolong tea produced in recent years in Taiwan has the sensory characteristics of a little sour, sweet, a umami taste, freshness, and an obvious floral flavor. A few GABA teas will have an obvious honey and dried fruit flavor. The roasted flavor and fire stink flavor is influenced by the roasting steps during the GABA Oolong tea making process. Some GABA teas with floral flavor do not have a roasted flavor because they are not roasted, while roasted GABA teas will have a varying degree of roasted flavor [14]. These special sensory attributes were perceived by the young Taiwanese consumers in this study.

Figure 4 and Figure 5 demonstrate the results of a correspondence analysis on the selected terms with a Chi-square distance calculation that describes the sensory characteristics of Taiwanese GABA Oolong tea as perceived by consumers in the first and the second test of CATA, respectively. In Figure 4, 26 sensory characteristics are selected to analyze. F1 explains 60.26% of variance, while F2 explains 26.22% of variance for all the data. In Figure 5, 24 sensory characteristics are selected to analyze. F1 explains 47.73% of variance, while F2 explains 18.93% of variance for all the data. The results of the correspondence analysis in Figure 4 show that the B-CY-P and B-NP-B1 tea samples, which scored the highest in terms of overall liking, had positive sensory characteristics such as floral flavor, sugary flavor, honey flavor, fruity flavor, late sweetness, smoothness, late fragrance, and softness. The B-CY-H sample, which scores the lowest in terms of overall liking, had negative sensory characteristics such as bitterness, astringency, and woody flavor. The results of the correspondence analysis in Figure 5 showed that the S-NT-L and B-NP-B2 tea samples, which scored the highest in terms of overall liking, had positive sensory characteristics such as brightness, sweetness, honey flavor, late sweetness, freshness, smoothness, and late fragrance. The B-NT-G2 sample, which score the lowest in terms of overall liking, had negative sensory characteristics such as sour, bitterness, astringency, and grassy/ Chinese herb flavor.

Samples with lower scores in terms of overall preference correlated more with negative terms. According to Figure 2 and Figure 3, the golden-yellow and light-yellow colors of tea liquid are placed in the second and fourth quadrants, respectively. This shows that customers perceived that a sample with more golden-yellow characteristics in its tea liquid had fewer light-yellow characteristics. In reality, however, the colors of the tea liquid of both types of GABA Oolong tea are paler, which shows that consumers cannot define clearly the golden-yellow and light-yellow colors of tea liquid, thereby validating the aforementioned deduction.

The results of CA for 12 different Taiwanese GABA Oolong tea in Figure 4 and Figure 5 are integrated into Figure 6 by PCA following the previous study [23]. PCA standardizes all the variables during the analysis, so it is possible to integrate the separate results of CATA analysis by different rounds of respondents. The GABA Oolong tea is put together for further comparison and discussion. In Figure 6, F1 explains 52.61% of variance, while F2 explains 22.41% of variance for all the data. It can be seen from Figure 6 that the overall liking for GABA Oolong tea is directly proportional to the *y*-axis. B-NP-B1, B-NP-B2, and B-CY-P with higher overall acceptability have sensory characteristics such as clearness, floral flavor, sugar flavor, honey flavor, fruit flavor, late sweetness, smoothness, lustrousness, late fragrances, and softness, and are distributed at the top of the PCA diagram. B-CY-H, B-NT-G1, and B-NT-G2 with lower overall acceptability have sensory characteristics such as sourness, bitterness, astringency, grassy/Chinese herb flavor, and woody flavor, and are distributed at the bottom of the PCA diagram.

PCA cannot accurately compare the differences among different samples. The respondents’ selection frequencies of various sensory attributes and overall liking judgements for 12 Taiwanese GABA Oolong teas were analyzed by PLSR and expressed as a correlation loading plot in Figure 7 refer to the results of previous studies [38]. The model was derived from attribute CATA data as the X-matrix and overall liking scores as the Y-matrix. If the different circles among the tea samples do not overlap, it means that the two tea samples are significantly different. It can be seen from Figure 7 that there are obvious differences between B-CY-P, B-CY-H, and other GABA Oolong tea samples, while there are less obvious differences between other tea samples.

Figure 8 shows the results of an evaluation that was conducted based on the partial least-squares (PLS) approach and concerns the sensory characteristics of GABA Oolong tea and its overall liking among consumers. The results show that the Taiwanese young consumers like the GABA Oolong teas with 13 specific sensory characteristics (brightness, lustrousness, sweetness, floral flavor, sugary flavor, honey flavor, fruity flavor, late sweetness, smoothness, freshness, late fragrance, coolness, and softness). However, consumers dislike the ones with another 6 specific sensory characteristics: sourness, saltiness, bitterness, woody flavor, roasted flavor, and grassy/Chinese herb flavor. The negative sensory characteristics of traditional GABA tea, such as fire stink flavor and sourness, will not appear after the improvement of GABA tea processing. Cheng’s (2016) study on commercially available packed tea showed that the bitterness, astringency, and bitter aftertaste of tea are the main factors behind a low consumer acceptability [48]. In a similar study, Lin (2010) pointed out that university students have a lukewarm response to strong tea because of its bitterness and astringency [49]. As shown in Figure 8, the larger gap in the terms that describe the lingering aftertaste of GABA Oolong tea result from the consumers’ inconsistent perceptions of these terms; terms that are liked by some are disliked by others. In addition, some consumers have a more intense feeling toward certain terms, while others were indifferent toward those terms. Therefore, the inconsistency of these terms among consumers could result in terms not commonly used in mainstream food culture.

## 4. Conclusions

The purpose of this study was to understand the acceptability of young Taiwanese consumers for GABA Oolong tea cold-brewed infusions. The sensory attributes of GABA Oolong teas were composed using the CATA method. The sensory qualities of Taiwanese teas were evaluated by a few experts through tea competitions. The award-winning teas, with high sensory quality and price, do not mean that these teas are liked by consumers. Our results are more indicative of consumer preferences and could provide scientific bases for the application of GABA Oolong tea by tea manufacturers and for their local and international promotion. We found that Taiwanese young consumers like the GABA Oolong teas with 13 specific sensory characteristics (brightness, lustrousness, sweetness, floral flavor, sugary flavor, honey flavor, fruity flavor, late sweetness, smoothness, freshness, late fragrance, coolness, and softness), but consumers dislike the ones with another 6 specific sensory characteristics (sourness, saltiness, bitterness, woody flavor, roasted flavor, and grassy/Chinese herb flavor). The terminology for describing the sensory characteristics of GABA Oolong tea was determined based on terms that were selected by over 20% of consumers, and a total of 21 major terms for describing Taiwan-made GABA Oolong tea were selected. Different processes of tea production will affect consumers’ preference for GABA Oolong tea. This study shows that young consumers in Taiwan can indeed accept the Taiwanese GABA Oolong tea because it already has the delightful sensory characteristics of traditional Oolong tea. Moreover, to know the sensory attributes of GABA tea samples liked by young Taiwanese consumers is important. It can help the manufacturer to produce GABA teas which young consumer will like, and can help the tea dealers to communicate with consumers in a language that is understood by young consumers in order to increase the sales. GABA Oolong tea beverage has a high sensory quality and additional healthcare effect, so the price of this kind of tea beverage can also be higher than that of ordinary tea beverages. The results of this study will be very helpful for the promotion of GABA tea beverage.

## Figures and Tables

**Figure 1 foods-11-02989-f001:**
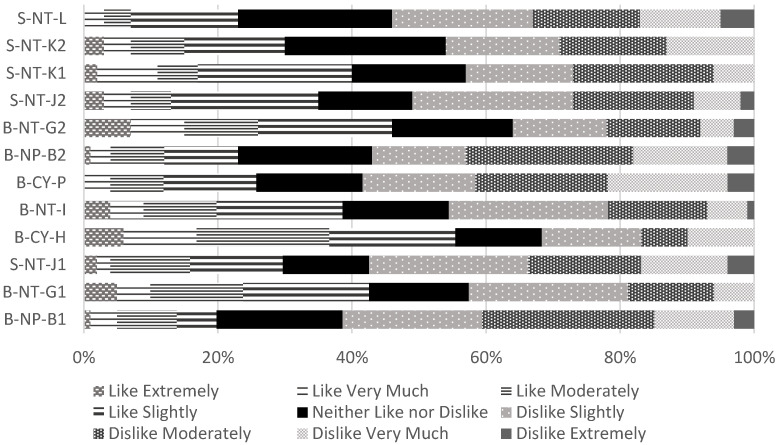
The consumers clustered by hierarchical clustering analysis of the overall liking mean values for 12 Taiwanese GABA Oolong tea made by the TTES #12.

**Figure 2 foods-11-02989-f002:**
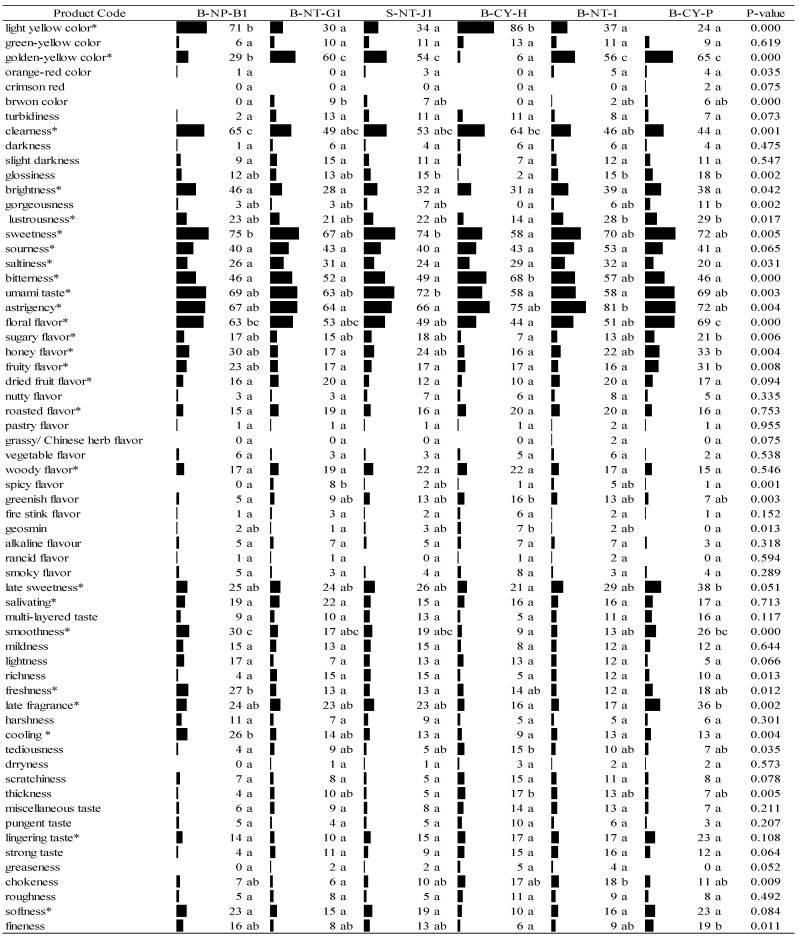
Participants’ selection frequencies of association for the cold infusions of the first round of Taiwanese GABA Oolong tea made by the TTES #12 based on the CATA method (*n* = 101). Significant difference (*p* < 0.05) of panelists’ selection proportions for each attribute among the six tea samples were determined by Cochran’s Q test. The pairwise post-hoc McNemar test (Bonferroni) was also performed, and the tea samples signed with different alphabet letters were significantly different in the frequencies of panelists’ selection for the attribute listed on the same row. Asterisks are marked on the attributes selected by more than 20% of respondents.

**Figure 3 foods-11-02989-f003:**
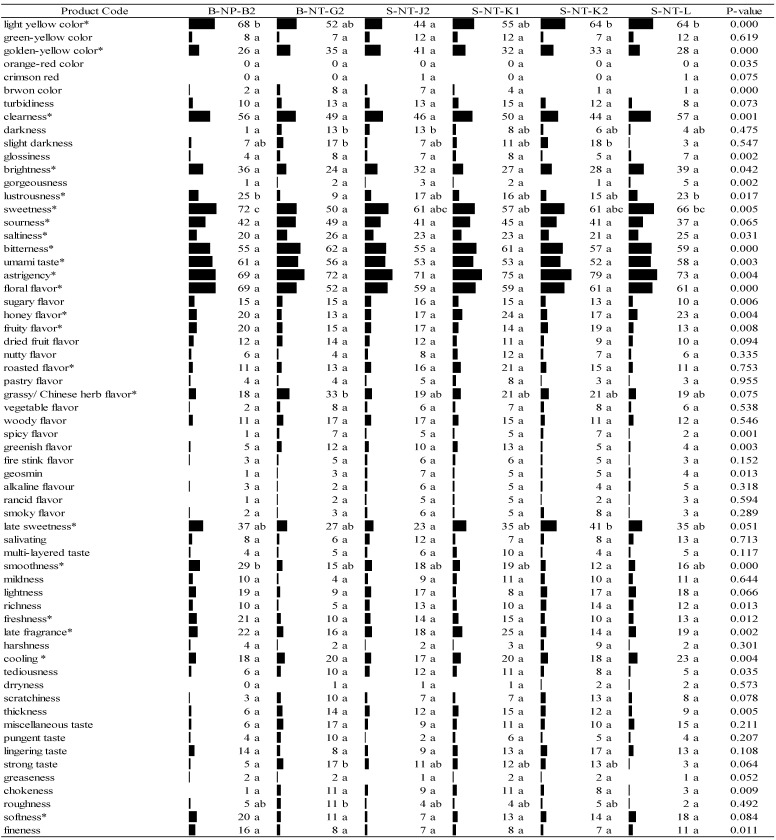
Participants’ selection frequencies of association for the cold infusions of the second round of Taiwanese GABA Oolong tea made by the TTES #12 based on the CATA method (*n* = 100). Significant difference (*p* < 0.05) of panelists’ selection proportions for each attribute among the six tea samples were determined by Cochran’s Q test. The pairwise post-hoc McNemar test (Bonferroni) was also performed, and the tea samples signed with different alphabet letters were significantly different in the frequencies of panelists’ selection for the attribute listed on the same row. Asterisks are marked on the attributes selected by more than 20% of respondents.

**Figure 4 foods-11-02989-f004:**
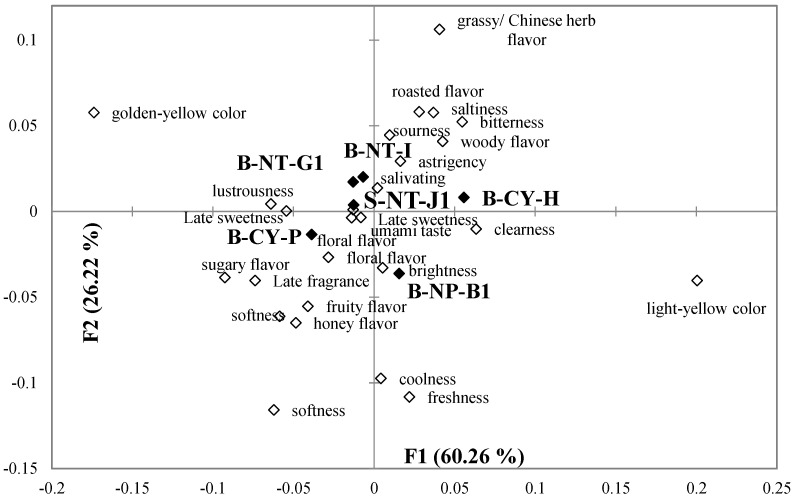
A bi-plot by correspondence analysis for the cold infusions of the first round of Taiwanese GABA Oolong tea made by the TTES #12 in association with the 27 sensory attributes selected by more than 20% of participants. Significant difference (*p* < 0.05) of panelists’ selection proportions for each attribute among the seven tea samples were determined by Cochran’s Q test. The pairwise post-hoc McNemar test (Bonferroni) was also performed, and the tea samples signed with different alphabet letters were significantly different in the frequencies of panelists’ selection for the attribute listed in the same row.

**Figure 5 foods-11-02989-f005:**
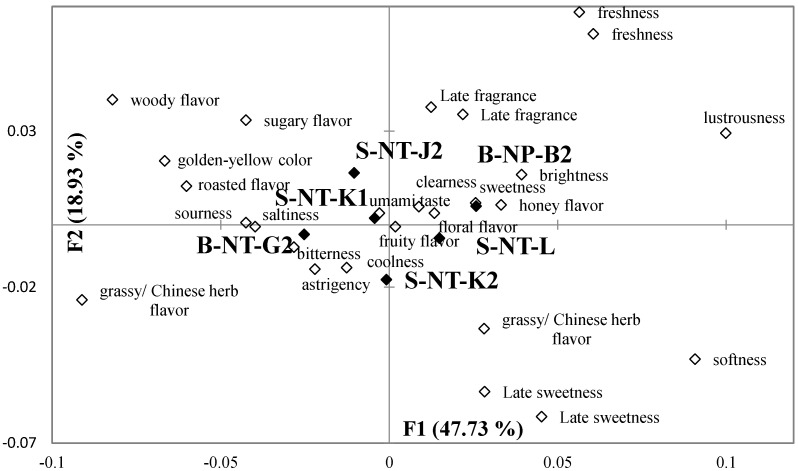
A bi-plot by correspondence analysis for the cold infusions of the second round of Taiwanese GABA Oolong tea made by the TTES #12 in association with the 24 sensory attributes selected by more than 20% of participants. Significant difference (*p* < 0.05) of panelists’ selection proportions for each attribute among the seven tea samples were determined by Cochran’s Q test. The pairwise post-hoc McNemar test (Bonferroni) was also performed, and the tea samples signed with different alphabet letters were significantly different in the frequencies of panelists’ selection for the attribute listed in the same row.

**Figure 6 foods-11-02989-f006:**
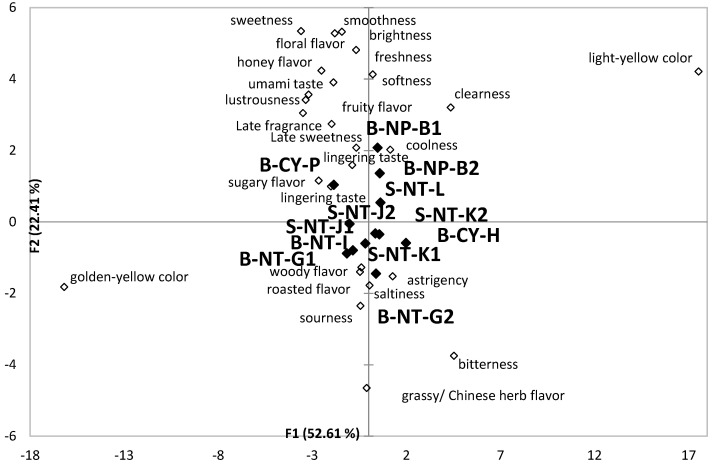
A bi-plot by PCA for the cold infusions of the 12 Taiwanese GABA Oolong tea made by the TTES #12 in association with the 21 sensory attributes selected by more than 20% of participants. Significant difference (*p* < 0.05) of panelists’ selection proportions for each attribute among the seven tea samples were determined by Cochran’s Q test. The pairwise post-hoc McNemar test (Bonferroni) was also performed, and the tea samples signed with different alphabet letters were significantly different in the frequencies of panelists’ selection for the attribute listed in the same row.

**Figure 7 foods-11-02989-f007:**
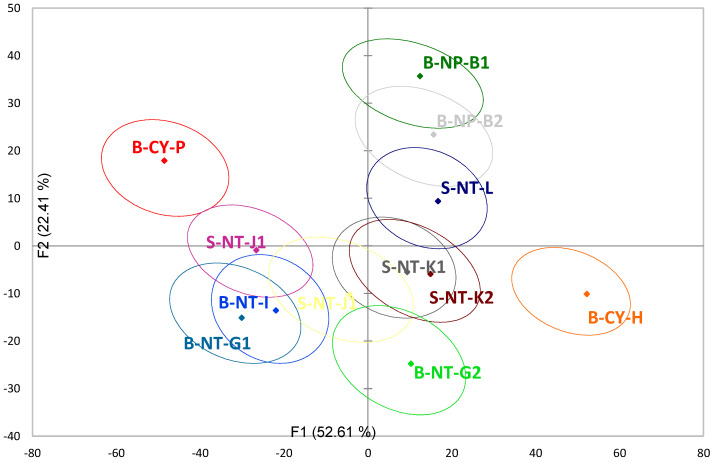
The partial least squares regression (PLSR) correlation loading plot of sensory, association, postprandial perception attributes, and overall liking of 12 Taiwanese GABA Oolong teas.

**Figure 8 foods-11-02989-f008:**
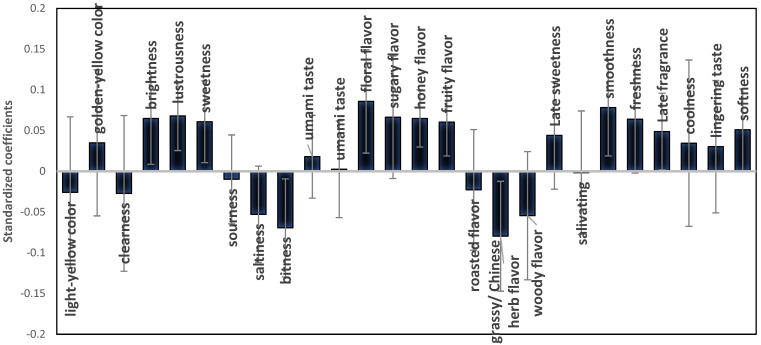
The standardized coefficient between sensory, association, postprandial perception attributes and overall liking of 12 Taiwanese GABA Oolong teas by partial least square (PLS) method.

**Table 1 foods-11-02989-t001:** The 12 Taiwanese GABA Oolong tea samples used in this study.

No	Product Code	Farming	Manufacturer	Classification of Dry Tea Leaves	GABA Content	Origin
1	B-NP-B1	Organic	B	Ball type	191	New Taipei City
2	B-NT-G1	Organic	G	Ball type	227	Nantou County
3	S-NT-J1	Conventional	J	Semi-ball type	178	Nantou County
4	B-CY-H	Conventional	H	Ball type	189	Chiayi County
5	B-NT-I	Conventional	I	Ball type	179	Nantou County
6	B-CY-P	Organic	P	Ball type	245	Chiayi County
7	B-NP-B2	Organic	B	Ball type	187	New Taipei City
8	B-NT-G2	Organic	G	Ball type	193	Nantou County
9	S-NT-J2	Conventional	J	Semi-ball type	189	Nantou County
10	S-NT-K1	Organic	K	Semi-ball type	192	Nantou County
11	S-NT-K2	Organic	K	Semi-ball type	165	Nantou County
12	S-NT-L	Conventional	L	Semi-ball type	225	Nantou County

The product code: the first code means the classification of dry tea leaves: B (Ball type), S (Semi-ball type); the second code means the origin of tea sample: NP (New Taipei City), NT (Nantou County), CY (Chiayi County); the third code means the manufacturer of tea sample: the letter means the first letter for the name of manufacturer.

**Table 2 foods-11-02989-t002:** The characteristics of consumers in this study (*n* = 101, 100 for 1st round and 2nd round).

Category		1st Round		2nd Round		Total
Sex	Male	43	42.57%	32	32.00%	37.31%
	Female	58	57.43%	68	68.00%	62.69%
Age	<20	7	6.93%	25	25.00%	15.92%
	20–29	69	68.32%	63	63.00%	65.67%
	30–39	6	5.94%	3	3.00%	4.48%
	40–49	2	1.98%	5	5.00%	3.48%
	50–59	12	11.88%	4	4.00%	7.96%
	>60	5	4.95%	0	0.00%	2.49%
Drinking tea habit	Habit of tea drinking	87	86.14%	87	87.00%	86.57%
	Heard about GABA tea	45	44.55%	45	45.00%	44.78%
	Drunk GABA tea	23	22.77%	16	16.00%	19.40%

**Table 3 foods-11-02989-t003:** The consumers acceptability for Taiwanese GABA Oolong tea made by TTES #12.

Sample	Overall Liking	Liking of Appearance	Liking of Flavor	Liking of Texture	Liking of Aftertaste
B-NP-B1	5.80 ^a^	6.04 ^ab^	5.85 ^ab^	5.80 ^a^	5.73 ^ab^
B-NT-G1	4.86 ^bcd^	5.98 ^ab^	5.14 ^bcd^	5.28 ^abc^	5.32 ^abcd^
S-NT-J1	5.60 ^ab^	6.24 ^ab^	5.70 ^ab^	5.76 ^a^	5.74 ^ab^
B-CY-H	4.44 ^d^	4.88 ^c^	4.61 ^d^	4.65 ^c^	4.65 ^d^
B-NT-I	5.04 ^abcd^	6.14 ^ab^	5.20 ^bcd^	5.39 ^abc^	5.29 ^bcd^
B-CY-P	5.84 ^a^	6.53 ^a^	6.02 ^a^	5.95 ^a^	6.09 ^a^
B-NP-B2	5.82 ^a^	6.12 ^ab^	5.87 ^ab^	5.81 ^a^	5.83 ^ab^
B-NT-G2	4.75 ^cd^	5.66 ^b^	4.84 ^cd^	4.92 ^bc^	4.90 ^cd^
S-NT-J2	5.31 ^abc^	5.94 ^ab^	5.29 ^abcd^	5.36 ^abc^	5.33 ^abcd^
S-NT-K1	5.06 ^abcd^	5.95 ^ab^	5.23 ^abcd^	5.32 ^abc^	5.37 ^abcd^
S-NT-K2	5.33 ^abc^	5.93 ^ab^	5.43 ^abc^	5.45 ^ab^	5.44 ^abcd^
S-NT-L	5.76 ^a^	6.17 ^ab^	5.70 ^ab^	5.81 ^a^	5.67 ^abc^

There is no significant difference marked with the same alphabet letters (*p* > 0.05).

## Data Availability

The datasets generated for this study are available on request to the corresponding author.

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
