# Peer review of "Understanding Young Taiwanese Consumers’ Acceptance, Sensory Profile, and Drivers of Liking for GABA Oolong Tea Beverages with Cold Infusions"

_foods, 2022, doi:10.3390/foods11192989_

Round 1

Reviewer 1 Report

The manuscript describes the consumer sensory perception of oolong tea that has high contents of GABA, which is supposedly good for health. Is the study supposed to be an advertisement for GABA oolong tea? The methodology has been poorly described. The results are poorly presented and are not very clear and lucid. Also, the discussion seems very superficial without much substance. The statistical analysis lacks proper rationale. There are numerous instances of sentences that are anecdotal and unscientific in nature. Also – sentence construction in many instances seem a bit shaky, although I am not the best judge for this purpose. Some specific comments are presented below –

Line 22-28 – These sentences should be revisited, and the authors should write maximum of 2 sentences that gives a clear rationale for the study.

Line 28-30 – The aim of the study does not state the objective of the study. Using couple sensory methods to collect data on 12 samples is not an objective of a research study. Why was the study done? To determine or understand what?

Line 30-31 – delete, redundant

Line 35 – popularity?

Line 37-38 – rephrase, and example of a sentence that poorly constructed

Line 40 – what is the purpose of repeating words/phrases in the Keywords that already appear in the title?

Line 75-79 – you have to be careful in writing about health benefits. You have to remember that most of the claims have not been proven beyond doubt. Also try to have more references to back such claims and put all the health benefits in one paragraph. Line 95-96 belongs here.

Line 92 – example of an unscientific and anecdotal sentence without a proper reference.

Line 96-100 – example of sentences that do not make much sense.

Line 105-108 – compare this with aim that you wrote in the abstract and you will understand what I meant.

Table 1 – “Product Code” and “Manufacturer y” – why are these important for this table. Is this something the reader needs to know? If yes – then explain it more, otherwise create your own codes. Why is there a line after the first 6 samples?

Line 132 – A 9-point hedonic scale (1 = Dislike Extremely; 5 = Neither like nor Dislike; 9 = Like Extremely) was used to determine the overall liking, liking of appearance, liking of flavor, liking of texture, and liking of aftertaste of the tea samples by the consumers [21 – the reference for hedonic scale is not required because it is used so widely].

Line 140 - …selecting was…

Line 141 – Taiwanese? Be consistent in your language

Line 145-146 – rephrase

Line 152 – why weren’t the consumers screened for product usage? Why 2 groups of consumers – explain the rationale behind it.

Table 2 – should be in the results section

Line 176 – why no Human Subjects Ethics approval?

Line 182 – what is sequential extinc Williams LS? Also – how was a Williams LS used in this design? The serving method is not clear. Was on 6 samples given? Were all the 12 samples tasted by all the participants in both the consumer groups? Is it usual to serve the infusions at ambient temperature? Specify the ambient temperature.

Line 193 – change ‘tick’ to ‘check’ throughout the manuscript

Line 200-202 – was it a temporal study? This sentence does not make sense.

Line 207 – shouldn’t this be MCA?

Line 211 – why not correlation matrix for PCA? What’s the rationale for using PCA with count/frequency data?

Line 215 – why PLSR? Liking data is on an interval scale while CATA data is frequency data. Penalty analysis might be better here. Otherwise give references where it was used.

Page 6 – could not make much sense of what results are presented here given the fact that it is not clear how the test was done with the 2 consumer panels. If each panel evaluated 6 different samples and the consumers did not taste all 12 samples and I don’t know if the comparisons are valid.

Table 3 – change the headings according to what I said for line 132

Figure 1 – How is this hierarchical clustering? Divide them into top 3 (7-9), middle 3 (4-6), and bottom 3 (1-3). Right now, it is not giving any meaningful information that can be discussed properly.

Line 283 – Figures 2 and 3 are not useful for readers at all. The authors can put them in supplementary information. They should find a way to just illustrate the important attributes only. An article should have tables and figures that helps understand the work and not make it more onerous on the readers. The readers will not go through such large amount of information in a figure to figure out the important attributes.

Line 310 – in my opinion, the authors could have just used the first group to select the important attributes of all the 12 infusions and then do the 2nd study with selected CATA terms and also dome liking part. The authors have mostly presented the frequency of terms without actually discussing the implications. They need to revamp this section.

Figures 4, 5 and 6 – 4 and 5 should be MCA I think. Since it is clear to me that the panels tasted 6 samples each, how was this data combined for the PCA because the authors said that the participants were different. I don’t think either PCA or the PLSR was the right choice here even if the authors are able to explain the data matrix used.

Line 476-479 – how can the authors say this when their overall liking scores doesn’t support it. Just by running a PLSR between quantitative data and frequency data is hardly the right way to interpret acceptability.

Line 498 – give limitations of your study

Line 511-513 – false assertion because the liking scored do not support it.

Line 515-516 – false assertion

Author Response

The manuscript describes the consumer sensory perception of oolong tea that has high contents of GABA, which is supposedly good for health. Is the study supposed to be an advertisement for GABA oolong tea? The methodology has been poorly described. The results are poorly presented and are not very clear and lucid. Also, the discussion seems very superficial without much substance. The statistical analysis lacks proper rationale. There are numerous instances of sentences that are anecdotal and unscientific in nature. Also – sentence construction in many instances seem a bit shaky, although I am not the best judge for this purpose. Some specific comments are presented below –

Thank you very much for reviewing our paper. The GABA tea has additional health benefits than other tea in Taiwan. But the traditional anaerobic GABA tea has strong pungent smell and was not liked by consumer. The new production method of GABA tea is developed after two decades of repeated improvements in Taiwan. The tea experts claim that the sensory characteristics of new GABA tea has significantly improved. The purpose of this study is to understand the acceptance and sensory profile of young Taiwanese consumers, not the advertisement for GABA oolong tea

Line 22-28 – These sentences should be revisited, and the authors should write maximum of 2 sentences that gives a clear rationale for the study.

Thank you very much for reviewing our paper. These sentences are revised in Line 22-28.

Line 28-30 – The aim of the study does not state the objective of the study. Using couple sensory methods to collect data on 12 samples is not an objective of a research study. Why was the study done? To determine or understand what?

Thank you very much for reviewing our paper. The aim of the study is revised in Line 29-31. The 12 samples are equally divided into two group and evaluated by two different groups of respondents. The reasons for using two groups of respondents are listed in line 166-169.

Line 30-31 – delete, redundant

Thank you very much for reviewing our paper. The sentences in Line 30-31 are deleted.

Line 35 – popularity?

Thank you very much for reviewing our paper. These sentences are revised in Line 33-35.

Line 37-38 – rephrase, and example of a sentence that poorly constructed

Thank you very much for reviewing our paper. These sentences are revised in Line 36-38.

Line 40 – what is the purpose of repeating words/phrases in the Keywords that already appear in the title?

Thank you very much for reviewing our paper. The keywords are modified as “GABA Oolong Tea; Cold Infusion; CATA; Young Consumer; Acceptance”.

Line 75-79 – you have to be careful in writing about health benefits. You have to remember that most of the claims have not been proven beyond doubt. Also try to have more references to back such claims and put all the health benefits in one paragraph. Line 95-96 belongs here.

Thank you very much for reviewing our paper. These sentences are revised in Line 75-83. We have added the related references to every effect. The sentences in Line 95-96 have also been moved to Line 80-83.

Line 92 – example of an unscientific and anecdotal sentence without a proper reference.

Thank you very much for reviewing our paper. We have added the reference for this sentence in Line 95-96.

Line 96-100 – example of sentences that do not make much sense.

Thank you very much for reviewing our paper. These sentences are revised in Line 92-106. We explain more clearly the reasons why these GABA Oolong teas were chosen for this study.

Line 105-108 – compare this with aim that you wrote in the abstract and you will understand what I meant.

Thank you very much for reviewing our paper. These sentences are revised in Line 107-114. And the aim of this study in abstract is also revised in Line 29-31

Table 1 – “Product Code” and “Manufacturer y” – why are these important for this table. Is this something the reader needs to know? If yes – then explain it more, otherwise create your own codes. Why is there a line after the first 6 samples?

Thank you very much for reviewing our paper. The “Manufacturer y” should be “Manufacturer”. We can compare whether the different type dry tea leaves, different origin of tea samples and different dry tea leaves affect the sensory characteristics and consumer acceptability of GABA Oolong tea. The explanation is added in Line 127-129. The line after the first 6 samples is used to separated the different group of respondents participated in this study. There are 7 groups of respondents (704 people) participated the evaluation of 41 GABA Oolong tea sample in the first Taiwan GABA tea competition in 2020. In this study, the first six sample were evaluated by 101 respondents, and the last six sample were evaluated by 100 different respondents. The related details are showed in Line 166-169 and Line 240-242.

Line 132 – A 9-point hedonic scale (1 = Dislike Extremely; 5 = Neither like nor Dislike; 9 = Like Extremely) was used to determine the overall liking, liking of appearance, liking of flavor, liking of texture, and liking of aftertaste of the tea samples by the consumers [21 – the reference for hedonic scale is not required because it is used so widely].

Thank you very much for reviewing our paper. The sentences are removed and modified in Line 143-144.

Line 140 - …selecting was…

Thank you very much for reviewing our paper. The word is ”selecting”, not ”selectin”. We have revised and modified the sentences in Line 149.

Line 141 – Taiwanese? Be consistent in your language

Thank you very much for reviewing our paper. The word is replaced by “Taiwanese” in Line 149.

Line 145-146 – rephrase

Thank you very much for reviewing our paper. These sentences are revised in Line 154-163.

Line 152 – why weren’t the consumers screened for product usage? Why 2 groups of consumers – explain the rationale behind it.

Thank you very much for reviewing our paper. We would like to investigate the acceptability and the sensory attributes perceived by consumers of GABA Oolong tea from the perspective of dietary behaviors and dietary culture. Moreover, most Taiwanese consumers have the habit of drinking tea, whether it is hot or cold brewed (tea beverage). Therefore, the consumers in this study did not need to be screened.

In order to avoid the fatigue of smell and taste of the respondents and to prevent the time of assessment being too long. 2 different groups of consumers was use in this study. An exceeded 100 respondents were selected to participate in the CATA test. It is evaluated from the point of view of statistical sampling distribution, and the resulting data were found to be sufficiently representative of the distribution of the population. And from the perspective of dietary culture, these two groups of consumers are representative of the dietary behaviors of young Taiwanese consumers. In addition, we avoid events that could affect the results by closely controlling the experimental method and the design of the questionnaire.

Table 2 – should be in the results section

Thank you very much for reviewing our paper. Table 2 and related discussion are remove to Line 216-223

Line 176 – why no Human Subjects Ethics approval?

Thank you very much for reviewing our paper. The data of this study were obtained from the first Taiwan GABA tea competition in 2020 in Taiwan, which is a commercial competition. So there in no Human Subjects Ethics should be approval. But all subjects gave their informed consent for inclusion before they participated in this study by completing the written form.

Line 182 – what is sequential extinc Williams LS? Also – how was a Williams LS used in this design? The serving method is not clear. Was on 6 samples given? Were all the 12 samples tasted by all the participants in both the consumer groups? Is it usual to serve the infusions at ambient temperature? Specify the ambient temperature.

Thank you very much for reviewing our paper. It should be “sequential monadic technique with William Latin square design”. It is modified in Line 180. It means the respondent will assess one tea at a time in order, and the order of the tea for assess is in accordance with Williams Latin square design

The 12 tea samples made by TTES #12 cultivar in this study were evaluate by different rounds of respondents. The different 2 rounds of test are designed by independent Williams Latin square design. Total number of the respondents for the first round of CATA test was 101, and another 100 respondents participated the second rounds of CATA. The respondents of the first round assess the first 6 GABA Oolong tea sample, and the respondents of the second round assess the other 6 tea sample.

The temperature of the infusion should be 4-7 oC, and it is revised in Line 176-177.

Line 193 – change ‘tick’ to ‘check’ throughout the manuscript

Thank you very much for reviewing our paper. The words “tick” are all replaced by the “check” in all the manuscript.

Line 200-202 – was it a temporal study? This sentence does not make sense.

Thank you very much for reviewing our paper. Here is a misuse of the wrong method, and it has been removed from the manuscript.

Line 207 – shouldn’t this be MCA?

Thank you very much for reviewing our paper. In this study, the respondents can perceive or not perceive for each characteristic is still considered as nonparametric statistical test for CATA test. And the numbers of respondents were not the same between 2 rounds of test. Therefore, we keep using Correspondence Analysis to analyze the results of each test.

Line 211 – why not correlation matrix for PCA? What’s the rationale for using PCA with count/frequency data?

Thank you very much for reviewing our paper. The resulting data were found to be sufficiently representative of the distribution of the population for both 2 rounds of test of CATA (exceeded 100 respondents for each test). We convert the data into a percentage of the number of people perceived for each characteristic of GABA Oolong tea and calculate it by weighting the data for PCA analysis. This method can solve the problem of inconsistent number of respondents in two rounds of sets. Therefore, PCA is used when two sets of data are combined for analysis.

Line 215 – why PLSR? Liking data is on an interval scale while CATA data is frequency data. Penalty analysis might be better here. Otherwise give references where it was used.

Thank you very much for reviewing our paper. The format of data of CATA is convert into percentage for PCA analysis. The converted data is used for PCA could be used in PLSR. Both Penalty analysis and combination of Correspondence Analysis, PCA, and PLSR are both valid methods that can be used to analyze the results of CATA. We follow the previous study using the combined statistical method [1], and the study by other scholars have also confirmed the feasibility of this method [2]. These discussions are also added in the manuscript in Line 456 and Line 481-482.

  1. Liou, B.K.; Jaw, Y.M.; Chuang, G.C.C.; Yau, N.N.J.; Zhuang, Z.Y.; Wang, L.F. Important Sensory, Association, and Postprandial Perception Attributes Influencing Young Taiwanese Consumers’ Acceptance for Taiwanese Specialty Teas. Foods 2020, 9, 100.
  2. Ares, G.; Antunez, L.; Bruzzone, F.; Vidal, L.L.; Gimenez, A.; Pineau, B.; Beresford, M.K.; Jin, D.; Paisley, A.G.; Chheang, S.L.; et al. Comparison of sensory product profiles generated by trained assessors and consumers using CATA questions: Four case studies with complex and/or simil ar samples. Food Quality and Preference 2015, 45, 75-86.

Page 6 – could not make much sense of what results are presented here given the fact that it is not clear how the test was done with the 2 consumer panels. If each panel evaluated 6 different samples and the consumers did not taste all 12 samples and I don’t know if the comparisons are valid.

Thank you very much for reviewing our paper. The 12 tea samples made by TTES #12 cultivar in this study were evaluate by different round of respondents. The data from these two groups of respondents are both sufficiently representative of the distribution of the population because of the exceed 100 respondents. Related discussions are added in Line 167-173.

Table 3 – change the headings according to what I said for line 132

Thank you very much for reviewing our paper. The headings are changed.

Figure 1 – How is this hierarchical clustering? Divide them into top 3 (7-9), middle 3 (4-6), and bottom 3 (1-3). Right now, it is not giving any meaningful information that can be discussed properly.

Thank you very much for reviewing our paper. If the original data is not compared at the same time, but only the average value of acceptability is used to investigate the difference between samples, there is a possibility of misjudging the actual acceptability of the GABA Oolong tea samples. We can clearly see from the results in Figure 1 that the percentage of consumers who have a positive comments of GABA tea. The result were showed in Line 293-294 and Line 296-299.

Line 283 – Figures 2 and 3 are not useful for readers at all. The authors can put them in supplementary information. They should find a way to just illustrate the important attributes only. An article should have tables and figures that helps understand the work and not make it more onerous on the readers. The readers will not go through such large amount of information in a figure to figure out the important attributes.

Thank you very much for reviewing our paper. All the multivariate statistical analysis is prone to misjudgment if the raw data is ignored in the discussion. The correct interpretation of PCA results needs to be supported by the result of descriptive statistics to avoid misinterpretation. The focus of this paper is to show the readers the sensory attributes that consumers can perceive from the GABA Oolong tea. The sensory profiles of the different GABA Oolong teas can also be seen in the bar graphs in Figure 2 and Figure 3.

We agree with your suggestion to delete Figures 2 and 3, but we still want to keep these two figures.

Line 310 – in my opinion, the authors could have just used the first group to select the important attributes of all the 12 infusions and then do the 2nd study with selected CATA terms and also dome liking part. The authors have mostly presented the frequency of terms without actually discussing the implications. They need to revamp this section.

Thank you very much for reviewing our paper. The 12 tea samples made by TTES #12 cultivar in this study were evaluate by different rounds of respondents. We want to understand the sensory characteristics of GABA Oolong tea and not just explore the quality of GABA tea. The discussion method you suggest is not applicable to this study.

The discussion of different sensory attributes of GABA Oolong tea have been done in Line 385-390, Line 393-395, and Line 399-407. The new related discussions are added in Line 418-425

Figures 4, 5 and 6 – 4 and 5 should be MCA I think. Since it is clear to me that the panels tasted 6 samples each, how was this data combined for the PCA because the authors said that the participants were different. I don’t think either PCA or the PLSR was the right choice here even if the authors are able to explain the data matrix used.

Thank you very much for reviewing our paper. The format of data of CATA is convert into percentage for PCA analysis. The converted data is used for PCA could be used in PLSR. The combination of Correspondence Analysis, PCA, and PLSR are used to analyze the results of CATA. We follow the previous study using the combined statistical method [1], and the study by other scholars have also confirmed the feasibility of this method [2]. These discussions are also added in the manuscript in Line 456 and Line 481-482.

  1. Liou, B.K.; Jaw, Y.M.; Chuang, G.C.C.; Yau, N.N.J.; Zhuang, Z.Y.; Wang, L.F. Important Sensory, Association, and Postprandial Perception Attributes Influencing Young Taiwanese Consumers’ Acceptance for Taiwanese Specialty Teas. Foods 2020, 9, 100.
  2. Ares, G.; Antunez, L.; Bruzzone, F.; Vidal, L.L.; Gimenez, A.; Pineau, B.; Beresford, M.K.; Jin, D.; Paisley, A.G.; Chheang, S.L.; et al. Comparison of sensory product profiles generated by trained assessors and consumers using CATA questions: Four case studies with complex and/or simil ar samples. Food Quality and Preference 2015, 45, 75-86.

Line 476-479 – how can the authors say this when their overall liking scores doesn’t support it. Just by running a PLSR between quantitative data and frequency data is hardly the right way to interpret acceptability.

Thank you very much for reviewing our paper. The sentences are revised in Line 509-514. They should be the consumer like the GABA Oolong teas with some attributes.

Line 498 – give limitations of your study

Thank you very much for reviewing our paper. The sentences are revised in Line 535-537.

Line 511-513 – false assertion because the liking scored do not support it.

Thank you very much for reviewing our paper. The sentences are revised in Line 550-551. “The different process of tea production will affect consumers' preference for GABA Oolong tea.”

Line 515-516 – false assertion

Thank you very much for reviewing our paper. The sentences are revised in Line 552-556.

Reviewer 2 Report

Dear authors, while this paper is quite interesting, some aspects are not very clear, it should be furter improved for possible publication.

The comments are:

1. The abstract presented mainly present background, method, implication and results, but conclusions were neglected.

2. In the introduction section, line 103-108, mention your contribution to academia. How does your aim and objectives if answered reduce the gap in academia?

3. in Section 2.1 Tea sample: Detail the selection criteria of the 12 tea samples Taiwan GABA

4. in 2.2 Questionnaire design. The section needs to be more depth analyzed. What were the questions? What objectives do they answer? Where did you get the questions from (references studies used?) These could be presented.

5. in Line 204-217: Mention the advantages and disadvantages of the statistical analyzes used

6. I found some typos and grammatical errors. English editing is necessary

Best regards

Author Response

  1. The abstract presented mainly present background, method, implication and results, but conclusions were neglected.

Thank you very much for reviewing our paper. The abstract is revised in Line 22-.37

  1. In the introduction section, line 103-108, mention your contribution to academia. How does your aim and objectives if answered reduce the gap in academia?

Thank you very much for reviewing our paper. The aim and objectives of this study is to understand the sensory profiles, driver of liking, and the acceptance of consumers for GABA Oolong tea among young Taiwanese consumers. We wish the results of these study will be a useful reference for the tea farmers and manufacturers to make the GABA Oolong tea which is liked by Taiwanese young consumers. In academia, we also wish that people can get to know this newly developed GABA Oolong tea in Taiwan and understand the young consumers' awareness of the sensory characteristics of GABA Oolong tea. The sentences are revised in Line 106-113.

  1. in Section 2.1 Tea sample: Detail the selection criteria of the 12 tea samples Taiwan GABA

Thank you very much for reviewing our paper. The sentences are revised in Line 116-121.

  1. in 2.2 Questionnaire design. The section needs to be more depth analyzed. What were the questions? What objectives do they answer? Where did you get the questions from (references studies used?) These could be presented.

Thank you very much for reviewing our paper. The objectives of questionnaire design are to understand the sensory profiles, driver of liking (by CATA test), and the acceptance of consumers (9-point hedonic scale) for GABA Oolong tea. Total 62 descriptive terms regarding appearance, basic gustatory and tactile, aroma, flavor, and aftertaste of tea infusions were adopted in the questionnaire of CATA. Total 62 descriptive terms regarding appearance, basic gustatory and tactile, aroma, flavor, and aftertaste of tea infusions were adopted in the questionnaire of CATA. These sentences are revised in Section 2.2.

  1. in Line 204-217: Mention the advantages and disadvantages of the statistical analyzes used

Thank you very much for reviewing our paper. We have added considerations in the selection of statistical analysis methods in this section

  1. I found some typos and grammatical errors. English editing is necessary

Thank you very much for reviewing our paper. We have revised the typos and grammatical errors in the manuscript.

Reviewer 3 Report

1- At the very beginning of the summary, what is CATA could be explained.

‘The check-all-that-apply (CATA) method is an appropriate, child-friendly approach to get insights on how young consumers perceive a food product and identify the most relevant sensory attributes that affect children's hedonic perception.’

https://onlinelibrary.wiley.com/doi/abs/10.1111/joss.12253#:~:text=The%20check%2Dall%2Dthat%2Dapply%20(CATA)%20method,that%20affect%20children's%20hedonic%20perception.

‘Check-all-that-apply (CATA) questions are versatile multiple choice questions which are being increasingly used for product sensory characterization with consumers.’

https://www.researchgate.net/publication/281220448_Check-all-that-apply_CATA_questions_with_consumers_in_practice_Experimental_considerations_and_impact_on_outcome

2- In the introduction to the summary it could be emphasized that consumer acceptance testing and CATA testing were the main design of the survey.

3- It was called GABA Oolong Tea Drinks in the title. It is not clear exactly what these are.

4- The introduction is also very confusing. In the introduction section; What is Taiwan GABA tea? How is it obtained? Are there any survey studies on this subject? The answers to the questions "What did you do, why did you do it" could have been given more clearly and fluently. It could also include information about what kind of tests were done and why you chose them. It was not specified why the competition was based.

5- The aim here is to what extent does the new generation know these teas? Determining whether they consume consciously or not. What is the goal could have been stated more clearly.

6- The difference and importance of GABA oolong tea could also be given. Literature information could be added in terms of health.

7- After all these were given, the survey and the results should be expressed, and who did what and what was found, if any, similar work should be given.

8- Taiwanese young consumers were mentioned, but in Table 2, those who participated in the survey were selected in the 20-60 age range.

9- The working title can be shorter and different.

‘Sensory Evaluation with CATA test on Taiwan GABA Oolong tea.’

Author Response

1- At the very beginning of the summary, what is CATA could be explained.

‘The check-all-that-apply (CATA) method is an appropriate, child-friendly approach to get insights on how young consumers perceive a food product and identify the most relevant sensory attributes that affect children's hedonic perception.’

‘Check-all-that-apply (CATA) questions are versatile multiple choice questions which are being increasingly used for product sensory characterization with consumers.’

Thank you very much for reviewing our paper. We have added the introduction of CATA in the Abstract.

2- In the introduction to the summary it could be emphasized that consumer acceptance testing and CATA testing were the main design of the survey.

Thank you very much for reviewing our paper. We have revised this part of manuscript in Line 106-113.

3- It was called GABA Oolong Tea Drinks in the title. It is not clear exactly what these are.

Thank you very much for reviewing our paper. The GABA teas is unique because the anoxic fermentation during tea production. It is mentioned in Line 54-58. The GABA Oolong tea is a kind of the GABA tea produced in Taiwan. t is mentioned in Line 65-71. The cold-brewed tea infusions were used as a beverage in Taiwan, and we used the cold-brewed tea infusions of GABA Oolong tea as a sample in this study. Therefore, “GABA Oolong Tea Beverages with Cold Infusion” is used in title of this study.

4- The introduction is also very confusing. In the introduction section; What is Taiwan GABA tea? How is it obtained? Are there any survey studies on this subject? The answers to the questions "What did you do, why did you do it" could have been given more clearly and fluently. It could also include information about what kind of tests were done and why you chose them. It was not specified why the competition was based.

Thank you very much for reviewing our paper. In this study, we hope to examine the acceptability and sensory characteristics of GABA tea to young consumers, rather than just defining the quality of tea from traditional tea competitions held annually in Taiwan. These sentences are revised in Line 63-65, Line 82-85, Line 91-100, Line 102-103, and Line 106-113.

5- The aim here is to what extent does the new generation know these teas? Determining whether they consume consciously or not. What is the goal could have been stated more clearly.

Thank you very much for reviewing our paper. The purpose of this study is revised in Line 106-113.

6- The difference and importance of GABA oolong tea could also be given. Literature information could be added in terms of health.

Thank you very much for reviewing our paper. The health care effects of GABA tea were added in Line 74-81.

7- After all these were given, the survey and the results should be expressed, and who did what and what was found, if any, similar work should be given.

Thank you very much for reviewing our paper. The findings of these study were revised in the section of conclusion in Line 537-561. The most important finding in this study is to sure that young consumers in Taiwan can indeed accept the Taiwan GABA Oolong tea. And we found that the Taiwanese young consumers like the GABA Oolong teas with 13 specific sensory characteristics. But the consumers dislike the ones with another 6 specific sensory characteristics. It can help the manufacturer to produce the GABA tea which young consumer will like, and can help the tea dealers to communicate with consumers with a language that is understood by young consumers in order to increase the sales.

8- Taiwanese young consumers were mentioned, but in Table 2, those who participated in the survey were selected in the 20-60 age range.

Thank you very much for reviewing our paper. According to the WHO definition, young people are defined as those aged between 18 and 29 years (Line 219-220). There are 75 % of young respondents in the first round and 85 % in the second round in this study. So the topic of this study is still set for Taiwanese young consumers.

9- The working title can be shorter and different.

‘Sensory Evaluation with CATA test on Taiwan GABA Oolong tea.’

Thank you very much for reviewing our paper. We will modify the working title of this study.

Round 2

Reviewer 1 Report

Line 165 – please use past tense through the manuscript – [There were seven groups respondents (total 704 participants) who participated…]. It is still not clear to me who tasted what in the two groups. Maybe refer back to table 1? This should be explained in the paper and not in the rebuttal. Only I am reading your rebuttal – not the readers.

Line 178 – the Williams Latin Square still does not make sense to me. Please describe how a LS design was used with 12 samples and more that 12 participants. Did you create multiple LSs?

Line 198 – explain how the stats was done in this section. Your readers should be able to understand what you have done.

PCA - You did not answer my question – why not correlation matrix? Also – I did not understand the data transformation you did. Again – the readers should be able to understand what you did.

Same with PLSR - writing rebuttals with self-citation doesn’t help. The Ares et al. paper did not use PLSR. They used PCA on the Descriptive data and CA on the CATA data and then they compared the two methods.

Author Response

Line 165 – please use past tense through the manuscript – [There were seven groups respondents (total 704 participants) who participated…]. It is still not clear to me who tasted what in the two groups. Maybe refer back to table 1? This should be explained in the paper and not in the rebuttal. Only I am reading your rebuttal – not the readers.

Thank you very much for reviewing our paper. The tense of Line 165 is changed as past tense. The tea samples which were drunk by the different groups of respondents were explained in Line 174-176. “The respondents of first round drank the first 6 tea samples of Table 1, and the respondents of second round drank the last 6 tea samples of Table 1.”

Line 178 – the Williams Latin Square still does not make sense to me. Please describe how a LS design was used with 12 samples and more that 12 participants. Did you create multiple LSs?

Thank you very much for reviewing our paper. The LS design used in this study explained in Line 184-188. “Each latin square consisted of 6 respondents and 6 samples. Before the experiment, the complete Williams Latin square for 120 respondents was designed by the software Com-pusense Cloud (Compusense, Inc., Guelph, Ontario, Canada). The sequence of tea samples for 96 respondents of separated 101 and 100 respondents in the first and second rounds used the duplicate 16 latin squares.”

The 16 complete sets and remaining 1 set that did not fully satisfy the Williams Latin square design used both of the 2 CATA test in this study. The remaining 1 set of respondents that did not fully satisfy the Williams Latin square design did not affect the results of other complete designed data statistically.

Line 198 – explain how the stats was done in this section. Your readers should be able to understand what you have done.

Thank you very much for reviewing our paper. The process of CATA and 9-point hedonic rating test are explained and modified in Line 203-206. “After selection the attributes for the one sample of CATA test was completed, the 9-point hedonic rating of this sample was assessed by the respondent. The questionnaire of CATA and 9-point hedonic rating tests were presented to the respondents through the software Compusense Cloud”.

The statistical methods for the CATA and 9-point hedonic rating test were originally described in Line 211-218 and Line 220-222, respectively. The statistical methods PLSR used between the CATA and 9-point hedonic rating test are revised in Line 222-225.

PCA - You did not answer my question – why not correlation matrix? Also – I did not understand the data transformation you did. Again – the readers should be able to understand what you did.

Thank you very much for reviewing our paper. The numbers of respondents are not the same between 2 rounds of test. Therefore, the covariance matrices with the transformed data from the different 2 rounds of CATA test have better effect for the PCA than the correlation matrix [1]. The explanation is showed in Line 218-220. “The converted data of the 2 rounds of CATA for 12 tea samples were analyzed by the co-variance model with covariance matrix for principal component analysis (PCA) to compare all the 12 samples at one time.”

The conversion method for the data of CA and calculation method used for PCA are explained and modified in Line 215-220. “The counts of the numbers of respondents perceived for each sensory attributes of GABA Oolong tea were further converted into a percentage of the perceived respondents to all respondents for each attribute. The converted data of the 2 rounds of CATA for 12 tea samples were analyzed by the co-variance model with covariance matrix for principal component analysis (PCA) to compare all the 12 samples at one time.”

  1. Borgognone, M.G.; Bussi, J.; Hough, G. Principal component analysis in sensory analysis: Covariance or correlation matrix? Food Quality and Preference 2001, 12, 323-326, doi:10.1016/S0950-3293(01)00017-9.

Same with PLSR - writing rebuttals with self-citation doesn’t help. The Ares et al. paper did not use PLSR. They used PCA on the Descriptive data and CA on the CATA data and then they compared the two methods.

Thank you very much for reviewing our paper. The 6 references about the PLSR used in CATA test are added in Line 225-227. The PLSR is more appropriate when the aim is to predict liking based on a series of sensory attributes [2]. The relative explanation is added in Line 227-228.

  1. Laureati, M.; Cattaneo, C.; Lavelli, V.; Bergamaschi, V.; Riso, P.; Pagliarini, E. Application of the check-all-that-apply method (CATA) to get insights on children's drivers of liking of fiber-enriched apple purees. Journal of Sensory Studies 2017, 32, e12253, doi:10.1111/JOSS.12253.
